# Spectral Library of European Pegmatites, Pegmatite Minerals and Pegmatite Host-Rocks – The GREENPEG Project Database

Joana Cardoso-Fernandes[1,2], Douglas Santos[1], Cátia Rodrigues de Almeida[1,2,3], Alexandre Lima[1,2], Ana C. Teodoro[1,2] and the GREENPEG project team[+]

[1]Department of Geosciences, Environment and Spatial Planning, Faculty of Sciences, University of Porto, Rua Campo Alegre, 4169-007 Porto, Portugal

[2]ICT (Institute of Earth Sciences) – Porto Pole (Portugal), Rua Campo Alegre, 4169-007 Porto, Portugal

[3]Centro de Investigação de Montanha (CIMO), Instituto Politécnico de Bragança (IPB), Campus de Santa Apolónia, 5300-253

[+] A full list of authors appears at the end of the paper.

*Correspondence to*: Joana Cardoso-Fernandes (joana.fernandes@fc.up.pt)

**Abstract.** The GREENPEG spectral database contains the spectral signature, obtained through reflectance spectroscopy studies, of European pegmatites and minerals, as well as their host rocks. Samples include Nb-Y-F (NYF) and Li-Cs-Ta (LCT)-type pegmatites and host rocks from pegmatite locations in Austria, Ireland, Norway, Portugal, and Spain. The database contains the reflectance spectra (raw and with continuum removed), sample photographs, and main absorption features automatically extracted by a self-proposed Python routine. Whenever possible, spectral mineralogy was interpreted based on the continuum-removed spectra. A detailed description of the database, its content and structure, the measuring instrument, and interoperability with Geographic Information Systems (GIS) is available in this database report. Moreover, examples of how the data can be used and interpreted are also provided. The advantages and added value of the presented dataset reside on its European scale with representative samples from pegmatites with distinct genesis, mineralogy, structure, and host rocks that can be used as a reference for pegmatite exploration at a global scale through satellite image processing, for example. The reported spectral mineral assemblages can also be of interest when considering resource estimation or ore processing. Thus, it is expected that this open dataset, available on the Zenodo platform https://doi.org/10.5281/zenodo.6518318 (Cardoso-Fernandes et al., 2022b), will be a reference for distinct types of users ranging from academia to industry.

## 1. Introduction

Hyperspectral data were considered for a long time as an expensive tool for detecting and mapping the earth's surface minerals. However, with the launch of the Earth Observing 1 (EO-1) satellite in November 2000, which included Hyperion, the first spaceborne imaging spectrometer, remote sensing research was given a new low-cost tool (Nikolakopoulos et al., 2007). In recent years, a new generation of hyperspectral satellites became available which can take the application of remote sensing data and techniques to a whole new level. PRISMA (*Precursore Iperspettrale della Missione Applicativa*) is an Italian Earth observation satellite launched in March 2019, with a spectral range of 400-2500 nm (Guarini et al., 2018), and EnMAP (Environmental Mapping and Analysis Program) is a German hyperspectral satellite launched in April 2022, with a spectral

range of 420-2450 nm (Chabrillat et al., 2022). Both satellite products have a spatial resolution of 30 meters and are expected to be used for a wide range of applications, including land use and land cover mapping, and mineral and resource exploration (Cardoso-Fernandes et al., 2022a; Lazzeri et al., 2021; Santos et al., 2022a; Schodlok et al., 2022; Vangi et al., 2021; Wocher

et al., 2022). To make such applications more successful, it is fundamental to provide researchers and other users with minerals and rocks reference spectra in an open, ready-to-use format.

A spectral database was built in the frame of the GREENPEG project (https://www.greenpeg.eu/), funded by the European Commission Horizon 2020, which aims to develop multi-method exploration toolsets for the identification of European, buried, small-scale (0.01-5 million $m^3$) pegmatite ore deposits of the Nb-Y-F (NYF) and Li-Cs-Ta (LCT) chemical types (Müller et

al., 2022a). Moreover, the GREENPEG project also aims to enhance European databases of petrophysical and reflectance properties, for example adding new data on the properties of pegmatites and their green raw materials, including their spectral signature obtained through reflectance spectroscopy studies. This database is publicly available on the Zenodo platform https://doi.org/10.5281/zenodo.6518319 (Cardoso-Fernandes et al., 2022b).

The main objectives of this work are to present (i) the spectral library composed under the GREENPEG project, (ii) the formats

in which the database is made available and the interoperability with Geographic Information Systems (GIS) software, (iii) how the spectral library is organized, and (iv) how the data can be used and interpreted.

Previous databases have been published concerning pegmatite exploration. Cardoso-Fernandes et al. (2021b) provided a dedicated spectral library only for the Fregeneda–Almendra Aplite–Pegmatite Field in Central Iberia. The samples corresponded to LCT pegmatites considered to be the result of fractional crystallization of peraluminous granites derived from

partial melting of highly peraluminous, calcium-poor, and phosphorus-rich metasedimentary rocks of Neoproterozoic age, during the Variscan orogeny (Roda-Robles et al., 2018). Despite its relevance to the state of the art, the previous database lacks spectral reflectance data for other types of pegmatites, with distinct mineralogy, textures, and genesis. Fabre et al. (2021) presented a broader database of laser-induced breakdown spectroscopy (LIBS) spectra from multiple locations around the world, such as the Fregenda–Almendra and Gonçalo pegmatite fields in the Iberian Peninsula, and also from Canada and

Brazil. Although this database comprised samples from distinct pegmatites, the main focus was on Li-minerals, and therefore samples from LCT pegmatites.

Taking this into account, the GREENPEG spectral reflectance database presents several advantages and high added value when compared with the already available datasets: (i) it is the first database of this kind built at a European scale; (ii) it includes samples from distinct pegmatites with different mineralogy, structure, host-rocks, and genesis (anatectic and granite-

related); and (iii) it is the first open database providing data on pegmatites of the NYF chemical type. Samples include LCT- and NYF-type pegmatites and host rocks from pegmatite locations in Austria, Ireland, Norway, Portugal, and Spain.

This extensive database provided in multi-formats represents, therefore, a high-quality dataset to be used by multiple users of different backgrounds, from the academia to the mining industry. For better comprehension of the dataset, representative spectra were selected as an example to demonstrate how the spectral mineralogy included in the database was interpreted. To

complement the metadata already provided within the spectral library (Cardoso-Fernandes et al., 2022b), this work presents in

detail its structure, content, the process of sample preparation and data acquisition, as well as useful tips on usability and accessibility of the data. The expected use of the spectral library is (i) to provide spectral properties of LCT and NYF pegmatites and possible host-rocks, namely the overall spectral behaviour, the most important absorption features and respective interpreted spectral mineralogy; and (ii) to serve as input for spectral band selection for satellite-based identification of outcropping pegmatites, as already shown in previous studies (Santos et al., 2022b). Moreover, the data collected is of high significance to potential users, since it demonstrates how the dominant spectral mineralogy can differentiate from the mineralogy identified by visual inspection of a hand sample. Users can leverage the curated dataset with raw, continuum removed and resampled spectra, extracted absorptions features and interpreted spectral mineralogy for pegmatite identification and mapping using remote sensing data and techniques.

## 1.1. Case studies

The GREENPEG project develops research at various scales in three European demonstration sites (Fig. 1): (i) Leinster (Ireland); (ii) Wolfsberg (Austria); and (iii) Tysfjord (Norway). Complementary prospective areas in Spain and Portugal were selected for testing the developed methodologies within the GREENPEG project, including the usefulness of reference spectral data (Müller et al., 2022a).

Pegmatites can be economically enriched in a variety of critical and other rare metals as well as industrial minerals or gemstones. Raw materials (minerals) of interest in pegmatites include (Müller et al., 2022a): quartz, ceramic feldspar, industrial mica, Li mica, columbite-tantalite, and beryl (in both LCT and NYF pegmatites); allanite and monazite (only in NYF); and spodumene, petalite, amblygonite, and pollucite (only in LCT).

The pegmatite field in the Tysfjord area (Norway) has an extension of about 20 km$^2$ with 22 known NYF dikes emplaced in granite-type rocks (Müller et al., 2022a). Two groups of pegmatites were mapped, both emplaced in the Tysfjord granite gneiss (Müller et al., 2022b). The first group consists of older Paleoproterozoic and metamorphosed pegmatites with lens- to cigar-shapes reaching up to 400 m in size, formed from residual melts of the hosting granitic gneiss; while the second group is composed of younger, smaller undeformed pegmatites (400-379 Ma) formed by anatexis due to partial melting of the granitic gneiss during late Caledonian events (Müller et al., 2022b). The pegmatite samples for the spectral library were collected in 14 distinct pegmatite locations (either *in situ* or from historical drill cores), but with an emphasis on the Håkonhals and Jennyhaugen pegmatites, both exposed in large open-pit areas allowing for detailed remote sensing studies to be carried out in the scope of the GREENPEG project (Santos et al., 2022b; Teodoro et al., 2021). These granitic pegmatites are known for their mineral diversity with 157 identified minerals, while the most common accessory minerals are allanite-(Ce), fergusonite-(Y), columbite-(Fe), beryl, various sulphides, and fluorite, besides the major minerals quartz, plagioclase, K-feldspar (variety 'amazonite'), and biotite (Müller et al., 2017).

The pegmatite field in the Leinster area in Ireland shows around 70 km$^2$ of extension and with 18 known LCT pegmatites emplaced in either Paleozoic granitic or metasedimentary rocks (Barros and Menuge, 2016; Müller et al., 2022a). The pegmatites are mainly hosted along a 3 km wide NE–SW regional structure along the eastern margin of the S-type Leinster

Batholith (emplaced around 400 Ma) known as the East Carlow Deformation Zone (Barros et al., 2020; Barros and Menuge,
2016). The unzoned to weakly zoned pegmatite dikes can range from a few meters to up ~20 m in thickness and are considered
to be of anatectic origin (Barros et al., 2020; Barros and Menuge, 2016). The pegmatites are characterised by the predominance
of spodumene associated with variable amounts of quartz, plagioclase, microcline and muscovite; while accessory minerals
include columbite-tantalite, cassiterite, F-apatite, lithiophilite, spessartine, beryl, schorl and pyrite (Barros et al., 2020). Due
to a lack of pegmatite outcrops, some samples were collected from boulders dispersed in agricultural fields, but most of the
samples come from a diamond drilling campaign conducted in the Moylisha area.

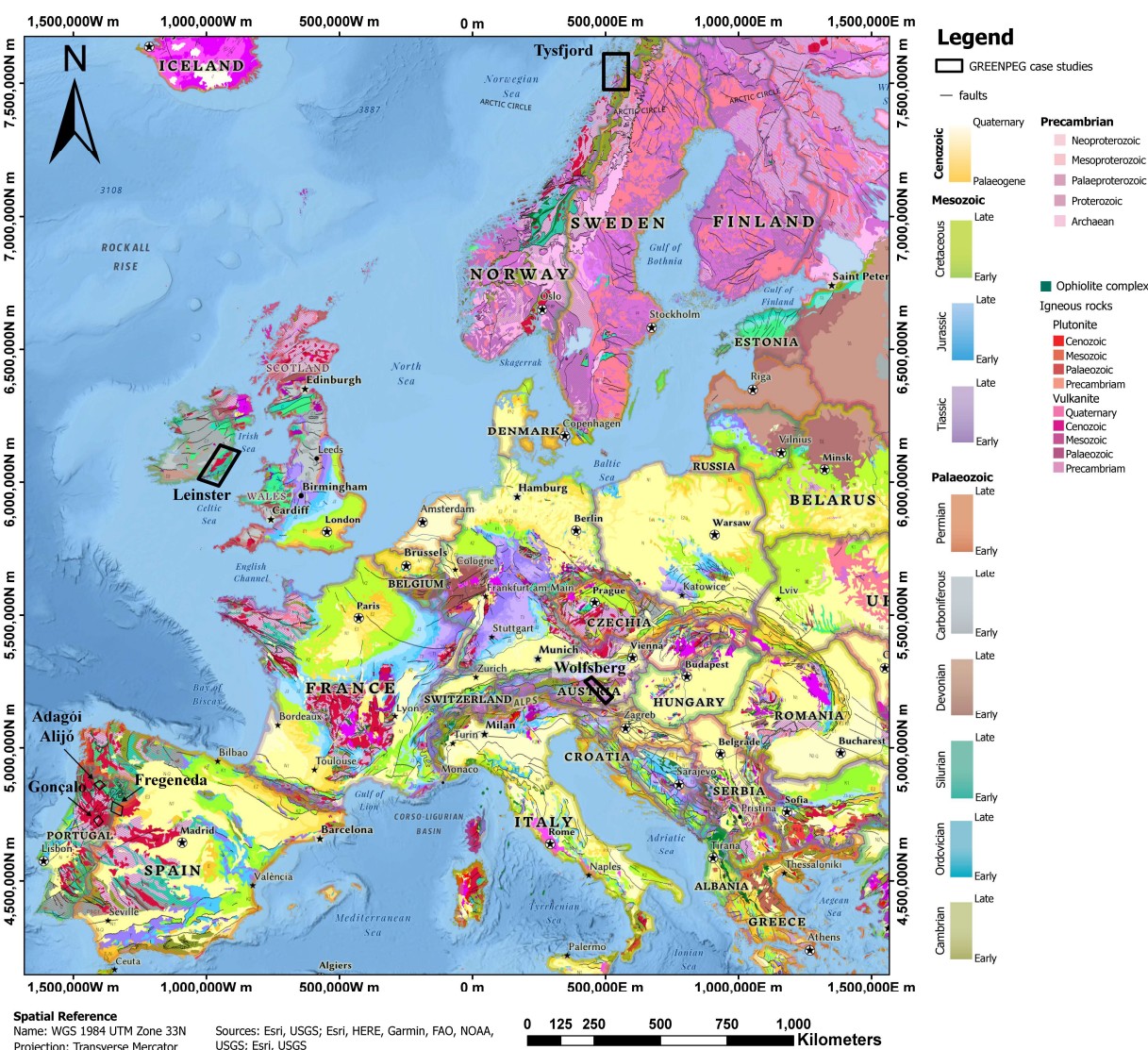

**Figure 1:** Location of the distinct sampled pegmatite fields in Norway (Tysfjord), Austria (Wolfsberg), Ireland (South Leinster), Portugal (Adagói, Alijó, Gonçalo), and Spain (Fregeneda) over the International Geological Map of Europe and Adjacent Areas (IGME 5000) at the scale 1:5,000,000 (adapted from Asch (2005) available at the European Geological Data Infrastructure (EGDI) platform -
https://www.europe-geology.eu/). Basemap provided by ESRI, NOAA, USGSS, FAO, Copyright © 1996-2022 Garmin Ltd., © 2022 HERE.

The Wolfsberg pegmatite field in Austria corresponds to an area of 25 km$^2$, where 14 known LCT dikes are emplaced in either amphibolite or mica-schist rocks (Gourcerol et al., 2019; Müller et al., 2022a). The LCT pegmatites are spatially associated with simple pegmatites and leucogranites formed during an extensional event in Permian times (Knoll et al., 2018). The pegmatites are mostly unzoned bodies (Göd, 1989). The minerals assemblage consists of K-feldspar, quartz, plagioclase, muscovite, garnet, tourmaline, spodumene, and other accessory phases such as beryl, apatite, cassiterite, or Nb-Ta phases (Knoll et al., 2018). Recent studies seem to indicate that the LCT pegmatites resulted from anatectic melts produced from the simple pegmatites and leucogranites, followed by fractionated crystallization (Knoll et al., 2018). Both pegmatites and leucogranites were overprinted during the Alpine orogeny with different intensities, affecting both structure and chemistry (Göd, 1989; Knoll et al., 2018). All samples analysed in this study were collected in recent drilling campaigns.

The samples from complementary test sites were collected in three distinct LCT pegmatite fields in Iberia, namely the Fregeneda-Almendra, Barroso-Alvão, and Gonçalo fields (Roda-Robles et al., 2018). All samples from Spain come from the Fregeneda side of the Fregeneda-Almendra pegmatite field (Roda-Robles et al., 1999; Vieira, 2010). The aforementioned pegmatite field spreads across a ~30 km-long and ~7 km-wide W–E belt of Neoproterozoic to Cambrian metasediments, with 11 distinct types of dykes identified, but with only five types of metasediment-hosted pegmatites worth mentioning, namely (Errandonea-Martin et al., 2022; Roda-Robles et al., 1999; Vieira et al., 2011): (i) concordant barren pegmatites; (ii) petalite-bearing pegmatites; (iii) spodumene-bearing pegmatites; (iv) lithium (Li)-mica pegmatites; and (v) spodumene+Li-mica pegmatites. The samples collected in Portugal are distributed among the remaining pegmatite fields. The Barroso-Alvão pegmatite field comprises both the Adagói and Alijó pegmatite sites. Martins (2009) defined distinct types of dykes in the region: (i) intragranitic pegmatites; (ii) quartz-andalusite dykes, (iii) barren pegmatites, (iv) spodumene pegmatites, (v) petalite pegmatites, and (vi) lepidolite pegmatites, all intruding metasedimentary rocks of Silurian-age. While in Alijó only spodumene crystals were identified, in Adagói both spodumene, petalite and eucryptite are found (Lima, 2000). Finally, the Gonçalo pegmatite field is characterised by granite-hosted sub-horizontal dykes with complex rhythmic layering with lepidolite-rich, albite-rich and quartz-rich layers, alternating with different textures (Neiva and Ramos, 2010; Roda-Robles et al., 2018). The aplite-pegmatite sills belong to the lepidolite and amblygonite subtypes, from the complex type of the LCT family of the rare-element class of pegmatites (Černý and Ercit, 2005).

In general, the LCT pegmatites from Iberia display different dips, from sub-vertical to sub-horizontal, reduced thicknesses (from <50 cm up to ≈30 m), and lengths usually lesser than 1 km (Roda-Robles et al., 2022). Moreover, these dykes do not present internal zoning, but show variable grain size, from aplitic to pegmatitic textures, with the biggest crystals being usually less than 12 cm long (Roda-Robles et al., 2022; Roda-Robles et al., 2018). Textures indicative of unidirectional solidification are commonly observed, mainly comb crystals of alkali feldspars and/or Li aluminosilicates, and/or a layering parallel to the contacts with the host rocks, while graphic textures or quartz cores are not commonly observed (Roda-Robles et al., 2022; Roda-Robles et al., 2018). Roda-Robles et al. (2018) proposed a petrogenetic link between peraluminous granites and the LCT aplite-pegmatite bodies of the Central Iberian Zone.

## 2.   Spectral library formats, structure and use

The spectral library is mainly focused on the spectral signature of the GREENPEG project demonstration sites: Norway (Tysfjord), Ireland (Leinster), and Austria (Wolfsberg). The spectral measurements were conducted on surface-collected and drill core samples provided by the GREENPEG project partners. Field campaigns allowed obtaining the spectral signature from representative samples from the test site in Portugal (Adagói). Additional samples from Portugal (Alijó, Gonçalo) and Spain (Fregeneda) previously collected were analysed and included in the spectral database. In order to ensure the representativity and completeness of the database, multiple samples from different parts of each pegmatite (including fresh and weathered regions) and samples from different pegmatites were collected. Drill core samples provided continuous exposure of pegmatite dykes allowing to assess the spatial distribution of mineral assemblages. In the case of zoned pegmatites, samples from each zone were collected. For further details on sample collection, please refer to Haase et al. (2022); (Haase and Pohl, 2022).

The database was originally created in a Microsoft Access database format (see chapter 2.2), but then converted to geodatabase/geopackage formats for interoperability with GIS software (see chapter 4), ArcGIS (Esri, Redlands, CA, USA) and QGIS, respectively. Original raw data was also resampled to match different satellite sensors. Therefore, users should: 1) Select the folder(s) of interest, 2) download the zip file, 3) extract the files, and 4) if necessary, follow the available tutorials.

The structure of the database is presented in Table 1. Each database level is separated according to the country of collection site: (i) Norway (Tysfjord), (ii) Austria (Wolfsberg), (iii) Ireland (Leinster), (iv) Portugal (Adagói, Alijó, Gonçalo), and (v) Spain (Fregeneda).

**Table 1:** Structure of the data available in the Zenodo database and respective content.

| Database Level | Folder Name | Content |
| --- | --- | --- |
| 0 | Database_files | Individual spectra and image files, stored by each demonstration site. |
| 1 | Microsoft_Access_database | Complete database with sample description and attachments. |
| 2 | Geodatabase | Geodatabase files to be displayed in ArcGIS. |
| 3 | Geopackage | Geopackage files to be displayed in QGIS, folders containing the attachment files to be linked to each geopackage file, and related tutorial. |
| 4 | Resampled_spectra | Individual spectra and image files resampled to match several satellite sensors' spectral resolutions (LC08 – Landsat 8, Sentinel-2, PRISMA, and WV3 – Worldview 3), stored by each demonstration site. |

## 2.1.   Sample preparation and data acquisition

All samples were dried at $\approx 50\,°C$ in a muffle furnace to remove any moisture from the sample that can influence the spectral behavior. The measurements were conducted in a dark room using the FieldSpec 4 (ASD Inc., Boulder, CO, USA) standard

resolution spectroradiometer covering the spectral range between 350 and 2500 nm with the following resolution: 3.0 nm @ 700 nm, 10.0 nm @ 1400 nm, and 10.0 nm @ 2100 nm (Fig. 2-a). A contact probe with an internal light source provided by a halogen bulb and a spot size of 10 mm was used for the laboratory measurements. Reflectance calibration was achieved with a Spectralon (Labsphere) plate with a maximum reflectance higher than 95% for the 250 to 2500 nm range and higher than 99% for the 400 to 1500 nm range. To increase the signal-to-noise ratio, each measurement comprises an average of 40 scans with four additional measurements acquired in each analysed spot that were later averaged into a final spectrum (Cardoso-Fernandes et al., 2021b) as demonstrated in Appendix A (Fig. A1). For each spot analysed, a description was made regarding the sample color and type of surface (fresh/weathered, exposed/freshly broken). The location of each measurement was annotated and photographed (Fig. 2-b). Taking into account the spatial variability of mineral assemblages within the pegmatite samples, several spots within the samples were measured to obtain representative spectra. Considering the spot size of 10 mm and the variable grain size within pegmatites, it is expected that in coarser-grained areas (pegmatitic texture) individual mineral spectra are obtained, while in fine-grained regions (aplitic texture) the spectra of each spot will represent a rock spectrum of the mineral assemblage within that spot.

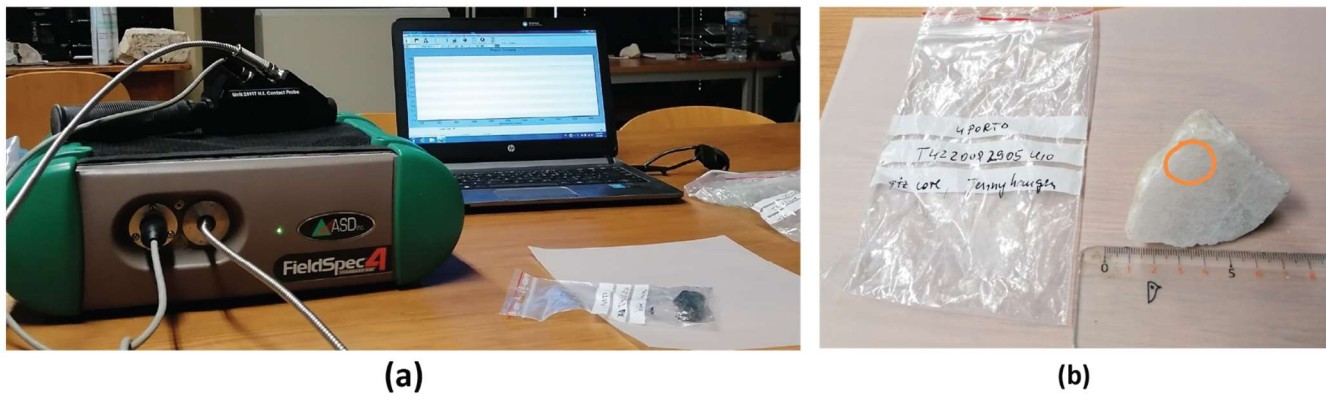

**Figure 2:** (a) FieldSpec 4 (ASD, Inc.) standard resolution spectroradiometer. (b) Example of an annotated photograph of the measurement spot in a sample from Tysfjord. A similar photograph accompanies each analysed spot in the spectral library.

For spectral post-processing, a self-proposed Python routine, using the *pysptools* library (Therien, 2013), was applied to eliminate the spectra continuum (normalisation) and to extract automatically the main absorption features and associated statistics (Cardoso-Fernandes et al., 2021b). To avoid the creation of artifacts and distortions, the continuum removal and respective feature statistics calculation were computed over the entire spectrum and over specific parts of the spectrum where the main absorption features are expected to occur, namely: OH (1350-1550 nm), water (1880-2040 nm), Al-OH (2160-2230 nm), Fe-OH (2230-2296 nm) and Mg-OH/$CO_3$ (2300-2370 nm). A comparison between performing the continuum removal over the entire spectrum and over specific parts of the spectrum is shown in Fig. A2.

However, absorption feature statistics such as the central wavelength position and the depth of one main absorption feature can be extracted in two ways: (i) in the first, the statistics are computed based on the channel with the minimum value in the continuum-removed feature; (ii) the second relies on the central wavelength of a quadratic function fitted to the band center

and one channel on each side. The band center from the quadratic function and the wavelength position of the band center channel may be very close in value when the feature contains a large number of channels. However, the quadratic function is less subject to noise in the spectrum since it uses more than one channel (Kokaly, 2011, 2008). Thus, in this study, the Python routine includes both statistics (one following the approach of continuum removal and the other based on the quadratic function fit method). The results between the two methods can be similar or show some differences from one case to another (see Fig. A3 and Table A1).

In the end, raw spectra were resampled to correspond to the spectral resolutions of multispectral and hyperspectral sensors, namely Landsat-8 Operational Land Imager (OLI), Sentinel-2 Multispectral Instrument (MSI), Worldview-3; and PRISMA. While there are potential limitations of resampling the laboratory spectra to match the spectral resolution of multispectral and hyperspectral satellite data, due to the loss of narrow and sharp diagnostic absorption features, resampling the spectral library can still provide valuable information for satellite image classification and lithological mapping (see Chapter 3).

## 2.2. Microsoft Access database

The database contains a table for each demonstration site and additional tables for the Portuguese and Spanish test sites. Each table follows the recommended nomenclature and content for hard rock samples defined in the GREENPEG Project Management Plan (Greenpeg D1.1, 2020), namely: (i) sample number (nr); (ii) sample description; (iii) locality; (iv) WGS84 Zone; (v) WGS84 Easting coordinate; (vi) WGS84 Northing coordinate; (vii) preparation; (viii) analysis; and (ix) place where the samples are stored (Fig. 3-a). The columns also provide information for each measurement: (i) a photograph of the sample; (ii) the observed sample color; (iii) the type of sample surface; (iv) the raw spectrum (as image and universal text file); (v) the normalised spectrum (either in an image and universal text file); (vi) the automatically identified absorption features; and (vii) the interpreted spectral mineralogy (Fig. 3-b) by comparing the acquired spectra to known reference spectra from the USGS spectral library and other published sources (Clark et al., 1990; Hunt, 1977; Pontual et al., 2008). A careful interpretation of the results was made in the context of the known mineralogy and geological setting of the area. Several database columns contain attachments (paper-clip icons) that can be previewed or saved to a local desktop. The number next to these icons indicates the number of available attachments (Fig. 3-b). Additionally, Microsoft Access automatically adds an ID column, where a primary key (number) is attributed to each entry in the database. The content of the database and the type of attachments is summarized in Table 2 while the detailed content is presented in Appendix A (Table A2).

All users should be aware that the sample description in the database was obtained by visual inspection and lithological or mineralogical identification based on the hand samples. Thus, at specific points on the rock or mineral sample, other spectrally active minerals may be present and contribute to the spectral signature captured at those locations and not just the mineral(s) identified in hand samples (see Section 2.2.1).

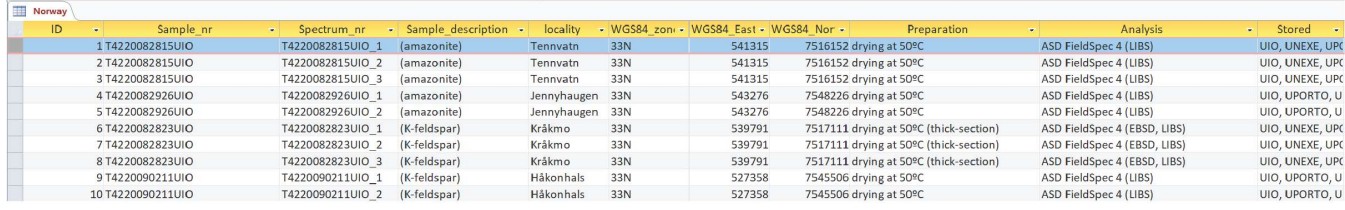

**(a)**

**(b)**

**Figure 3:** Database table preview for the Tysfjord demonstration site: (a) recommended content of hard rock sample list and (b) content specifically related to the spectral database (© Microsoft 2022).

**Table 2:** Content of the database in brief.

| Field | Description | Attachments files |
|---|---|---|
| Sample number (nr) | Sample identification following GREENPEG's nomenclature | – |
| Spectrum number | Spectrum number (sample number + number of the analysed spot within the sample) | – |
| Sample description | Description of the sample (provided by the partners or taken in the field by the authors) | – |
| Locality | Place where the samples were collected | – |
| WGS84 Zone | UTM zone | – |
| WGS84 Easting | X-coordinate in UTM (Easting) | – |
| WGS84 Northing | Y-coordinate in UTM (Northing) | – |
| Preparation | Sample preparation (for the spectral library and other parallel studies, the latter between brackets) | – |
| Analysis | Analytical methods employed (complementary studies are between brackets) | – |
| Stored | Where the sample and respective duplicates are stored (names represent the project partners) | – |
| Face color | The sample color in the measured spot | – |
| Face type | The sample face type in the measured spot | – |
| Photo | The sample photograph (measured spots are highlighted) | .png/.jpg |

| Raw spectra | The raw spectra (either in an image and universal text file) | .txt/.pdf |
| Processed spectra | The continuum removed spectra (either in an image and universal text file) | .txt/.pdf |
| Spectra absorptions | The automatically identified absorption features | .png/.csv |
| Spectral mineralogy | Interpreted spectral mineralogy | – |

### 2.2.1. Data interpretation and use

The identified spectral mineralogy of the investigated samples was achieved through the comparison of the measured
absorption spectrum with reference material spectra of the United State Geological Survey (USGS) spectral library (Clark, 1999; Clark et al., 2003; Kokaly et al., 2017) and other published libraries (Clark et al., 1990; Hunt, 1977; Pontual et al., 2008). The interpretation was done in the continuum-removed spectra by looking at the shape, symmetry, depth and wavelength position of the main absorption features. Following the steps proposed by Pontual et al. (2008): (i) the deepest absorption feature in the 2050-2450 nm region was identified as well as the corresponding mineral spectral group; (ii) within the spectral
group, the reference spectra that closest match the acquired spectra will represent the dominant spectral mineral; (iii) as spectral mixtures are often observed, other absorption features can be used to identify other minerals. For specific details on the spectral interpretation of pegmatite minerals, please refer to Cardoso-Fernandes et al. (2021a) and references therein.

Our results show that the spectral mineralogy identified does not necessarily match the minerals identified by observation of hand specimens and optical microscopy. This is because some silicates do not present necessarily diagnostic absorption
features (Spatz, 1997) or because the spectra are dominated by alteration minerals that are spectrally very active due to the presence of water/hydroxyl group and superimpose unaltered mineral domains (Hunt and Ashley, 1979). It is noteworthy that some of these clay minerals can dominate the spectra even in the most preserved, fresh samples as demonstrated by Cardoso-Fernandes et al. (2021). However, according to previously mentioned study, the absorption depth of these alteration minerals appears to correlate with the degree of alteration of the analysed mineral.

The representative reflectance spectra stored in the libraries can be utilised for satellite image processing, namely in the image classification tasks. To do so, the acquired spectra can be resampled to match the satellite sensors' spectral resolution and used as a target for algorithm training instead of the image pixels. This approach is, for example, an alternative satellite image processing method for the Leinster and Wolfsberg demonstration sites, where there are insufficient outcrops of pegmatite in the target area to serve as training areas in the satellite image processing, thus preventing the application of machine learning
algorithms (Greenpeg D2.3, 2021). Moreover, by comparing the location of the reflectance anomalies of the reference spectra with the location of the satellite sensor bands, it is possible to select the most indicative bands for image processing and improve the previously employed approaches (Santos et al., 2022b). Thus, the spectral library will be extremely helpful for the processing of Worldview-3 images, in particular, due to the increased spatial and spectral resolution.

### 3. Representative spectra of the demonstration sites

This section selects representative spectra of each demonstration site to demonstrate how the data available in the database can be interpreted with some examples of the spectra figure files.

Spectra of pegmatite samples from Tysfjord (Norway) are characterised by biotite/chlorite features mixed with crystallographic water, microfluid inclusions and alteration minerals, such as montmorillonite and illite (Fig. 4-a, -b). Granitic gneiss samples are characterised by strong $Fe^{2+}$ ramp-like absorptions caused by biotite/chlorite mixed with white mica, illite and/or

montmorillonite (Fig. 4-c, -d).

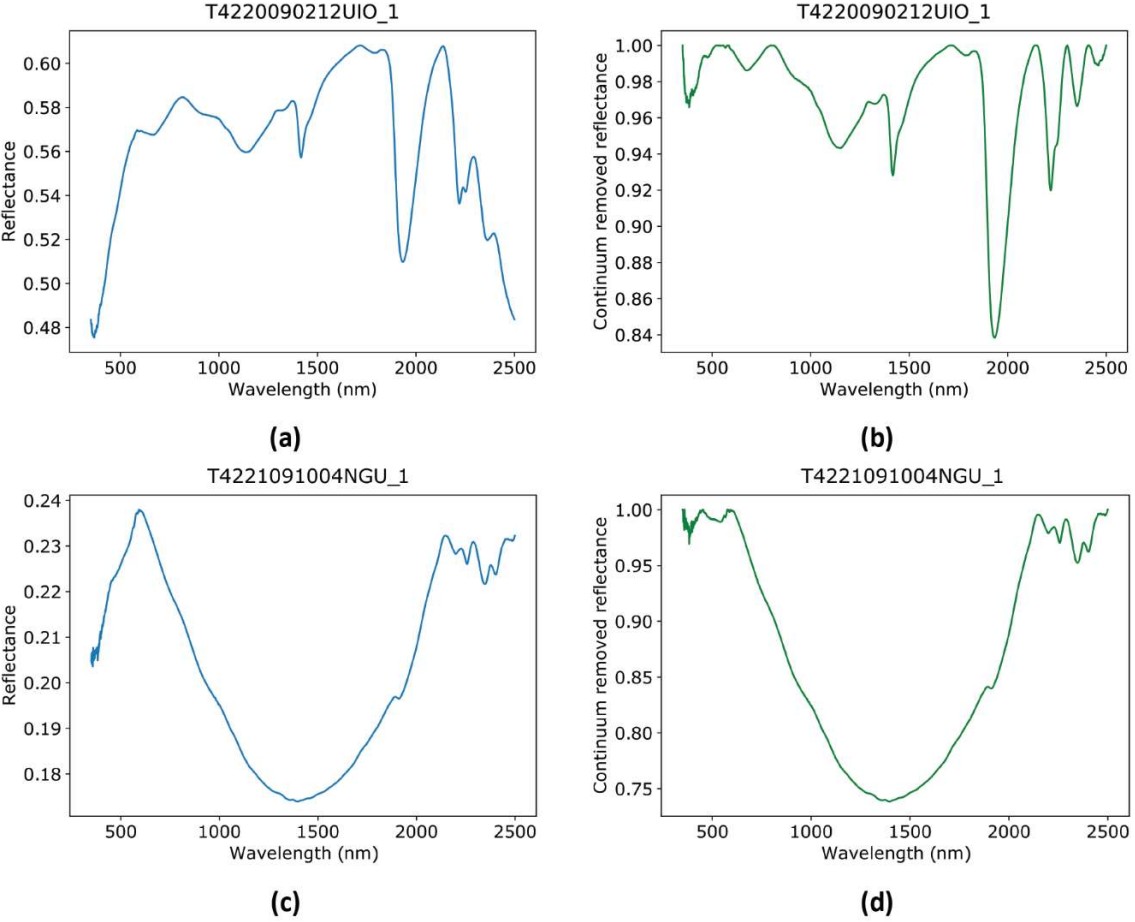

**Figure 4:** Representative spectra of samples from the Tysfjord demonstration site. Raw (a) and normalised (b) spectra of pegmatitic plagioclase showing features of illite mixed with montmorillonite and biotite and crystallographic water and/or water of microfluid inclusions. Raw (c) and normalised (d) spectra of granitic gneiss displaying a strong ramp-like $Fe^{2+}$ absorption and chlorite/biotite mixed
with white mica.

The spectra of pegmatite samples from Leinster (Ireland) are dominated by illite and/or montmorillonite features (Fig. 5-a, -b), sometimes mixed with white mica or orthoclase. Granite samples are characterised by $Fe^{2+}$ ramp-like absorptions caused by biotite/chlorite mixed with white mica, illite and/or montmorillonite (Fig. 5-c, -d). The schist samples have spectral features

of white mica or hydrated white mica, probably sericite, mixed with chlorite/biotite (Fig. 5-e, -f). Altered samples are

dominated by alteration minerals, such as illite or montmorillonite.

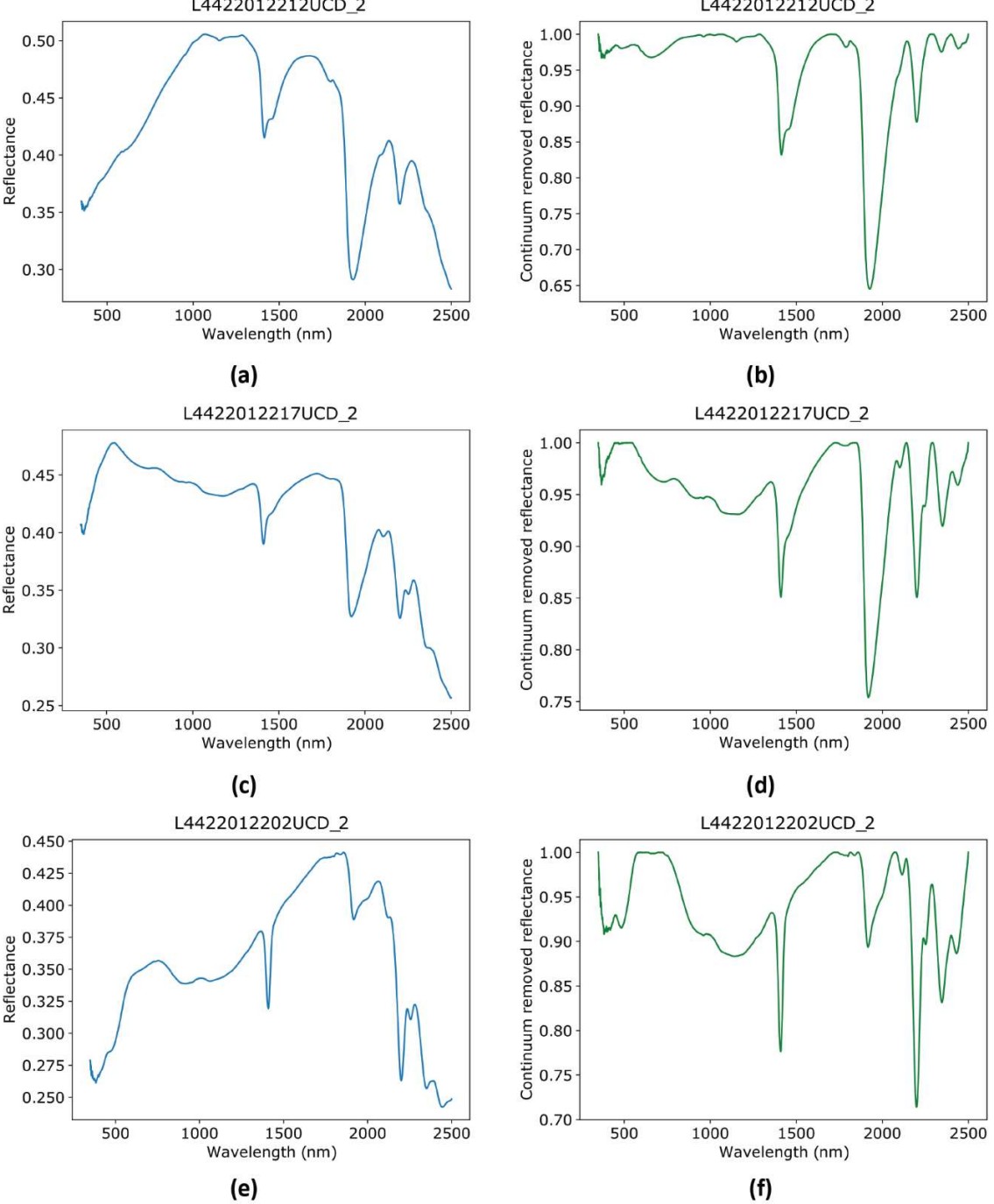

**Figure 5:** Representative spectra of samples from the Leinster demonstration site. Raw (a) and normalised (b) spectra of an albitised pegmatite sample showing spectral features of montmorillonite mixed with white mica/illite. Raw (c) and normalised (d) spectra of a granite

sample showing a ramp-like $Fe^{2+}$ absorption and features of biotite mixed with chlorite, illite and montmorillonite. Raw (e) and normalised (f) spectra of a schist sample showing spectral features of hydrated white mica, probably sericite mixed with chlorite/biotite.

The spectra of pegmatite samples from Wolfsberg (Austria) are dominated by white mica or hydrated white mica (probably sericite) sometimes mixed with illite or montmorillonite (Fig. 6-a, -b). The pegmatite host rock samples display features diagnostic for white mica, chlorite, biotite, phlogopite and/or carbonates, such as siderite and magnesite (Fig. 6-c, -d).

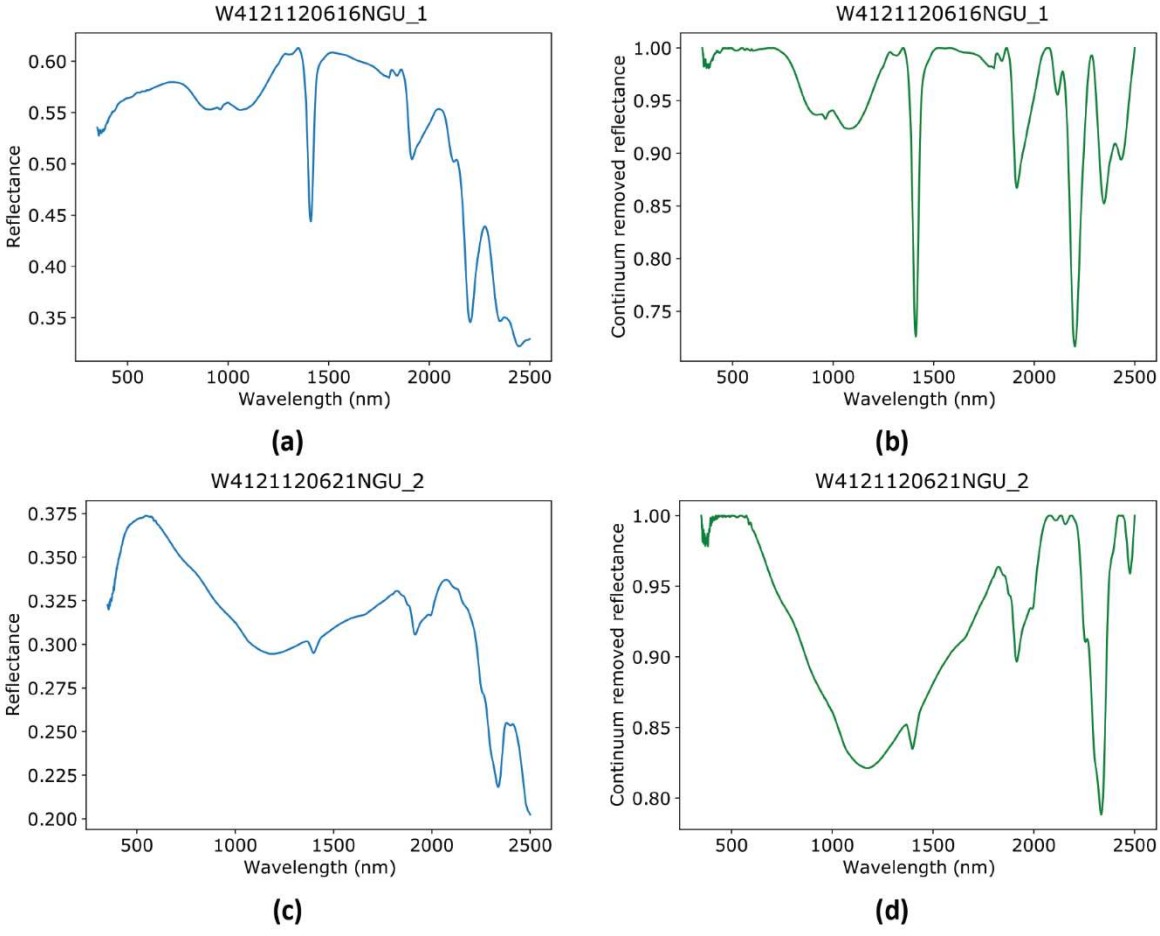

**Figure 6:** Representative spectra of samples from the Wolfsberg demonstration site. Raw (a) and normalised (b) spectra of a pegmatite sample showing spectral features of hydrated white mica, probably sericite. Raw (c) and normalised (d) spectra of the pegmatite-host rock (probable amphibolite) sample showing ramp-like $Fe^{2+}$ absorption spectral features diagnostic for carbonates (siderite) possibly mixed with chlorite.

While it is clear that the overall spectral signature of LCT pegmatites is mostly associated with alteration minerals such as clays, as observed in previous studies (Cardoso-Fernandes et al., 2021a), the spectral behaviour of NYF pegmatites is more dominated by biotite/chlorite features. Since biotite and chlorite (to the exception of cookeite) are not expected to appear in LCT pegmatites, the characteristic Fe and Mg features allow to spectrally discriminate between NYF and LCT pegmatites. Moreover, although not identified in this study, diagnostic, sharp absorption features of Rare Earth Elements (REE) in the

visible spectrum (~400 to 700 nm), caused by f→f transitions of the 4f electrons (Möller and Williams-Jones, 2018; Rowan et al., 1986; White, 1967), could be observed associated with allanite or monazite mineral phases. However, if these minerals are not present in NYF samples, distinguishing different types of pegmatite solely through spectral features and mineral identification can be challenging.

The superposition of the satellite bands over the reference spectra or the resampled spectra to match the satellite sensors' resolutions is very useful to select the most crucial bands for image processing through band ratios, RGB combinations or image classification was demonstrated by previous research works (Nikolakopoulos and Papoulis, 2016; Santos et al., 2022b). To assess the potential use of resampled spectra, Fig. 7 shows the spectra of pegmatite and host rocks from Wolfsberg according to different multi- and hyperspectral sensors. Sentinel-2 (Fig. 7-b) shows the lowest discrimination between the host rocks and pegmatite, followed by Landsat-8 (Fig. 7-a), Worldview 3 (Fig. 7-c) and PRISMA (Fig. 7-d) with the closest spectral range of the laboratory spectroradiometer. In the case of Landsat-8 (Fig. 7-a), Band 6 (1.57-1.65 µm) is crucial for lithological discrimination with pegmatite showing a reflectance peak in that range that is not observed in the host rocks. The higher spectral resolution of Wordview-3 (Fig. 7-c) allows greater discrimination with the AlOH absorption feature well marked for the pegmatite within the SWIR-6 band (2185-2225 nm), while the host rocks only show the carbonate absorption within the SWIR-8 band (2295-2365 nm).

Nonetheless, it should be noted that the acquired spectra only cover the visible and near-infrared (VNIR) and shortwave infrared (SWIR) regions which provide useful information for the identification of clays, micas, and alteration minerals in pegmatites. For a complete characterization of pegmatites and their major rock-forming minerals (i.e., quartz and feldspar), a sensor covering the thermal region would be needed since such silicates only present diagnostic absorption features in that range (Spatz, 1997). However, most satellite data products do not cover this thermal region either, or when they do, the spatial resolution is too coarse to be of use for pegmatite identification. Thus, the spectra in the GREENPEG database are representative of the signature captured by satellite sensors. Careful use and interpretation of the spectra must be made by any potential user since it is established that the pegmatite dominant spectral mineralogy might not be exclusive of pegmatite rocks (Cardoso-Fernandes et al., 2021a).

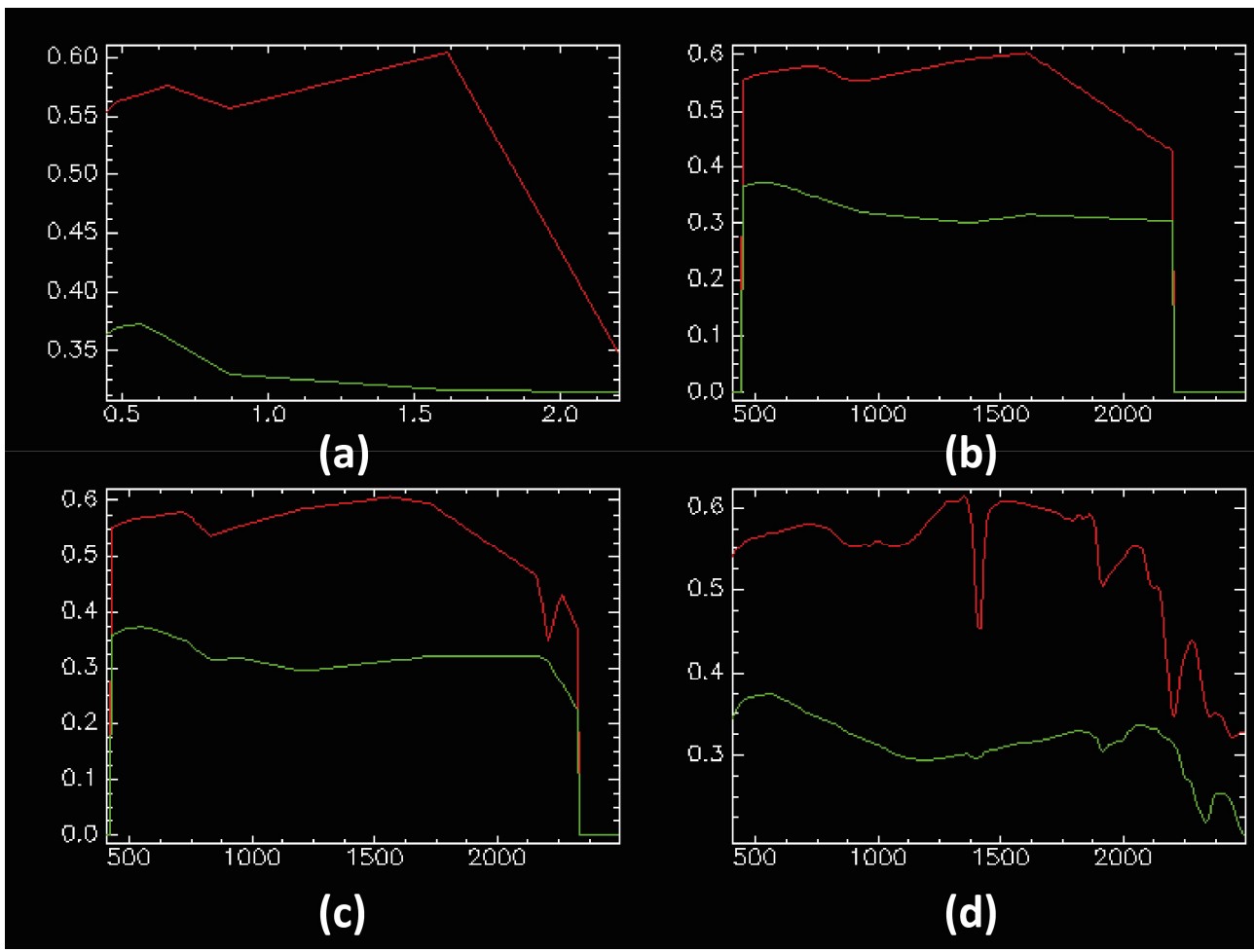

**Figure 7.** Comparison of the pegmatite (red) and host-rock (green) spectra from Wolfsberg (Fig. 6) resampled to match the spectral resolution of different satellite sensors: (a) Landsat-8 Operational Land Imager (OLI); (b) Sentinel-2 Multispectral Instrument (MSI); (c) Worldview-3; and (d) PRISMA. X-axis units are in micrometres (a) and nanometres (b-d).

## 4. Interoperability with GIS

Interoperability of the spectral database with a GIS environment is crucial for the successful exploitation of the spectral library and established dataset. Therefore, the spectral database is provided in two additional formats: a geodatabase file format (.gdb) and a geopackage file format (.gpk). The geodatabase file can be opened in ArcMap or ArcGIS Pro (Esri, Redlands, CA, USA) and the attachments are automatically previewed or opened since they are stored within the geodatabase (Fig. 8). The information from the spectral library, including attachments, can be easily previewed using (i) the 'Explore' tool, which opens a pop-up window (left panel of Fig. 8), or (ii) by accessing the 'Attributes' tab (right panel of Fig. 8). In this panel, the user can double-click each file to open them in the system default program according to the file type.

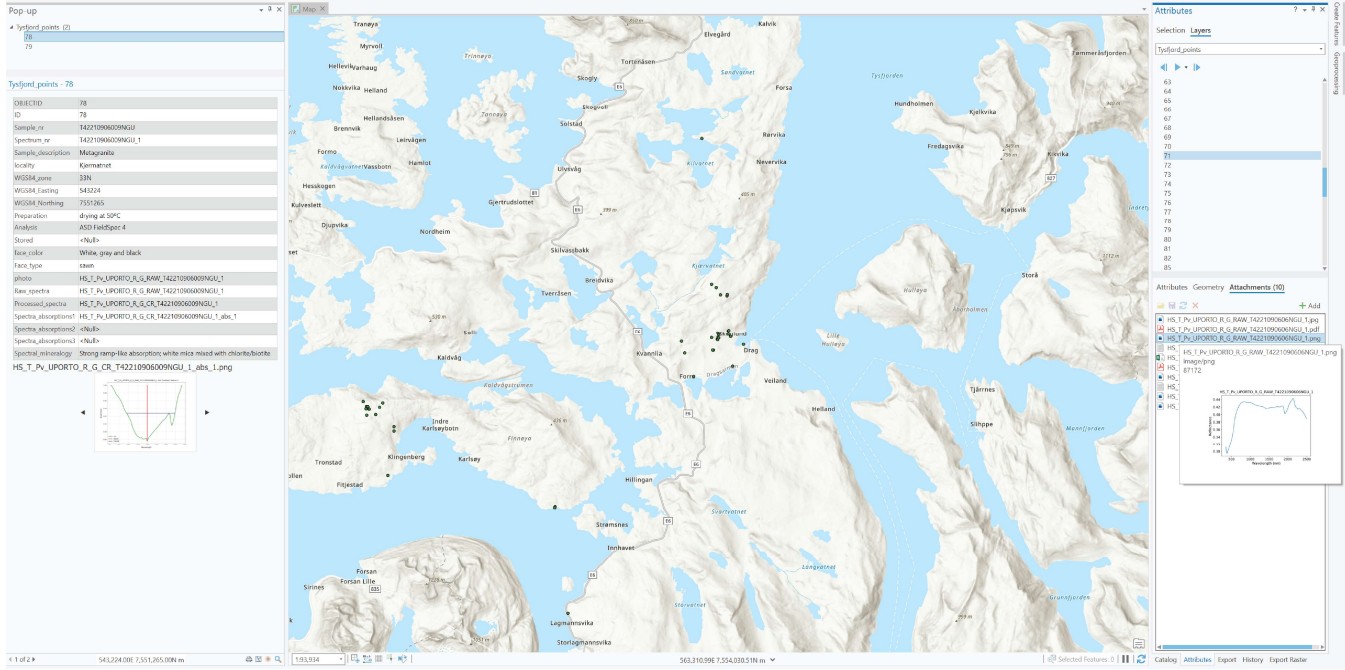

**Figure 8:** Illustration of how to use the geodatabase in ArcGIS Pro on the example of the Tysfjord demonstration site: the 'Explore' tool is shown on the left panel and the 'Attributes' tab in the right panel. Basemap provided by ESRI.

For QGIS users, the processing is manual and for that, the files are stored externally. For this approach, the geopackage file is
provided together with a folder with related attachments. The QGIS user needs to re-link the attachments in the folder with the
paths stored in the columns of the layer's attribute table of the geopackage file. A tutorial for this latter step is also provided
within the respective database level together with the geopackage file. After linking the files, the attachments can be previewed
in QGIS (Fig. 9). The information from the spectral library, including attachments, can be easily previewed using the 'Identify'
tool that opens a pop-up window (right panels of Fig. 9). It should be noticed that unlike with the geodatabase, the geopackage
file only allows previewing image files (.png, .jpeg, etc.).

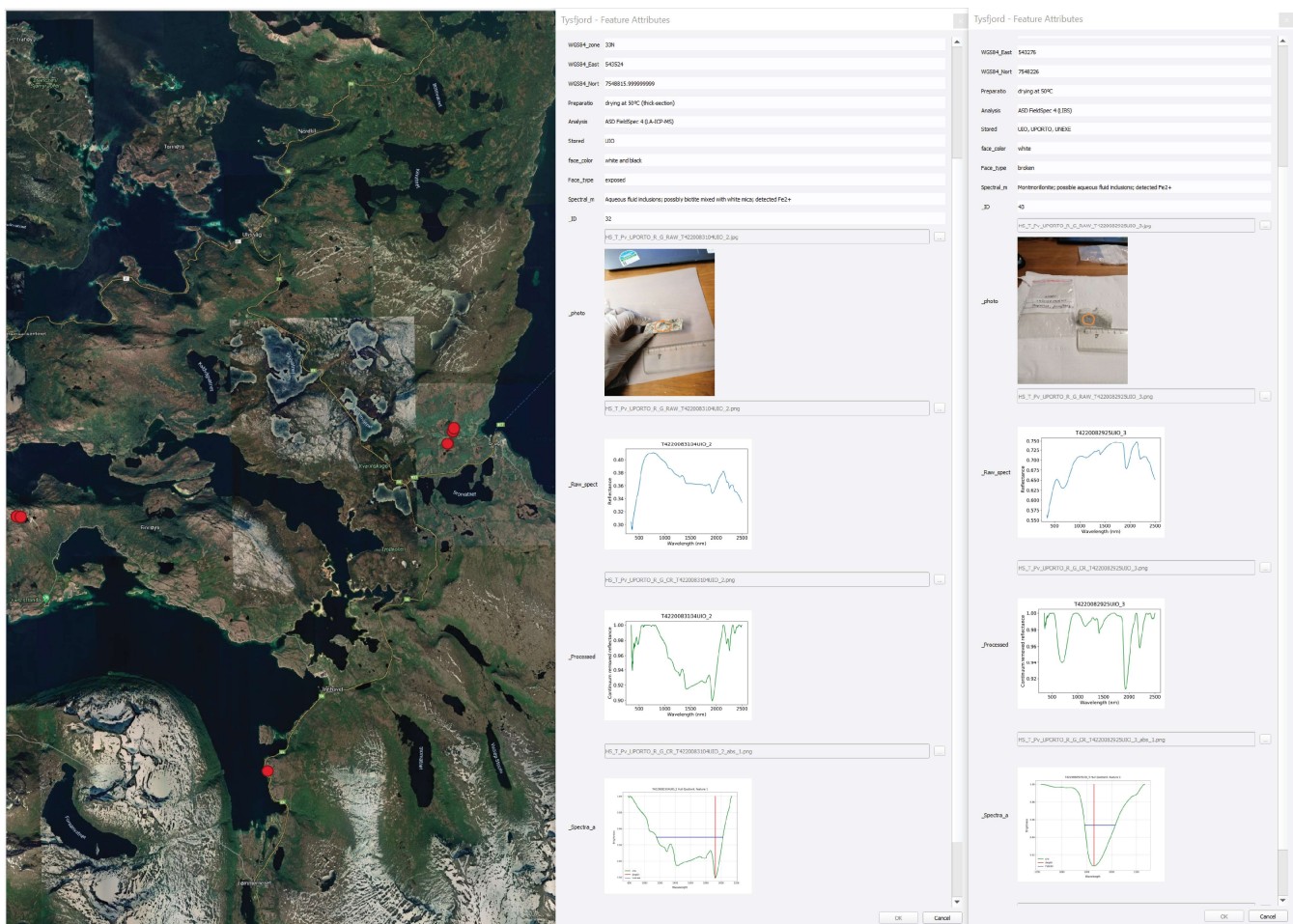

**Figure 9:** Illustration of how to use the geopackage file in QGIS on the example of the Tysfjord demonstration site: the 'Identify' tool opens a pop-up window with the attachments (right panels). Basemap provided by © Google Earth 2022.

## 5. Data availability

Since the GREENPEG project is committed to an Open Research Data Pilot (ORDP), the spectral library is openly available on the Zenodo platform (https://www.zenodo.org/communities/greenpeg-project) with the DOI identifier https://doi.org/10.5281/zenodo.6518318 (Cardoso-Fernandes et al., 2022b), following the GREENEPG ORDP (Greenpeg D8.1, 2020). All formats (Microsoft Access, geodatabase, geopackage) and individual spectral library files are stored as previously described for easy accessibility and use. File naming and metadata records follow the rules established in the Project

Management Plan and Data Management Plan (Greenpeg D1.1, 2020; Greenpeg D8.1, 2020). However, the spectral library version available in the Zenodo platform has confidential and sensitive information redacted, as requested by GREENPEG's industry partners.

## 6. Code availability

The original Python routine is available as supplementary material to the work of Cardoso-Fernandes et al. (2021b).

## 7. Conclusions

The GREENPEG European spectral database presented in this work aims to add new data on the properties of pegmatites and their green raw materials as well as different host rocks, allowing the evaluation of the potential for discriminating the two. The advantages and added value of the presented dataset reside on its European scale, therefore presenting reference spectra for pegmatites of both NYF and LCT chemical types, and also representative samples from pegmatites with distinct genesis, 355 mineralogy, structure, and host rocks.

This spectral database is also relevant because the results show that the spectral mineralogy identified does not necessarily match the minerals identified by observation of hand specimens and optical microscopy. Such information is crucial for users trying to detect other pegmatites worldwide. The reported spectral mineral assemblages can also be of interest when considering resource estimation or ore processing due to the large information provided for the distinct pegmatites sampled.

Since this dataset is aimed to be used as a reference for pegmatite exploration at a global scale, and distinct types of users are expected to benefit from its content, data usability and accessibility were a priority, thus ensuring a multi-format database and clear interoperability with a GIS environment to fully exploit the data provided.

The reflectance spectra stored in the database can: (i) enable accurate identification and characterization of pegmatites, aiding in understanding their formation and evolution; (ii) facilitate comparative analysis of pegmatites from different locations, 365 providing insights into regional variations due to the library's comprehensive collection of spectra; (iii) be utilised for satellite image processing and image classification in the early stages of pegmatite exploration. Currently, the spectral library is being used for the processing of Worldview-3 images over the Tysford (Norway) case study.

Overall, this library empowers researchers and professionals to better understand, characterize, and map pegmatites, contributing to advancements in geology, mineralogy, and resource exploration.

 **8.  Appendices**

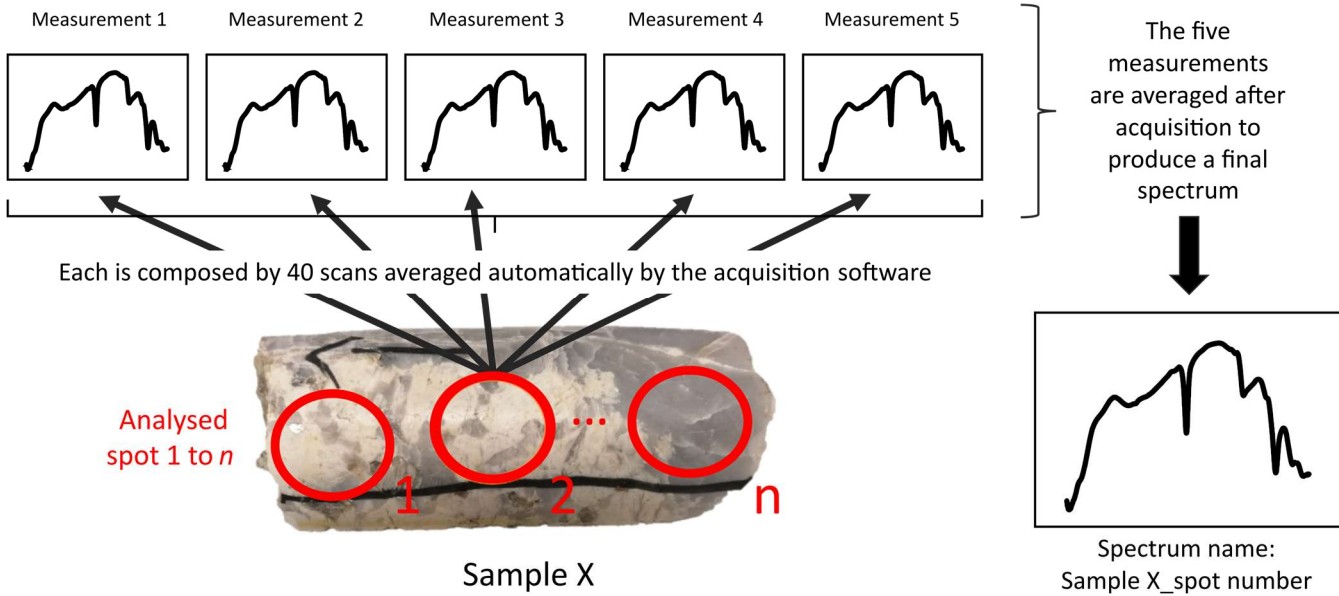

**Figure A1:** Schematic representation of the spectra acquisition process: several spots are analysed within one sample; for each spot, five measurements are done in a row; each measurement is the automatic average of 40 scans; a final spectrum is created by averaging the five measurements and the spectra name contains the sample name and the corresponding stop number to discriminate from other spectra acquired in other spots within the same sample.

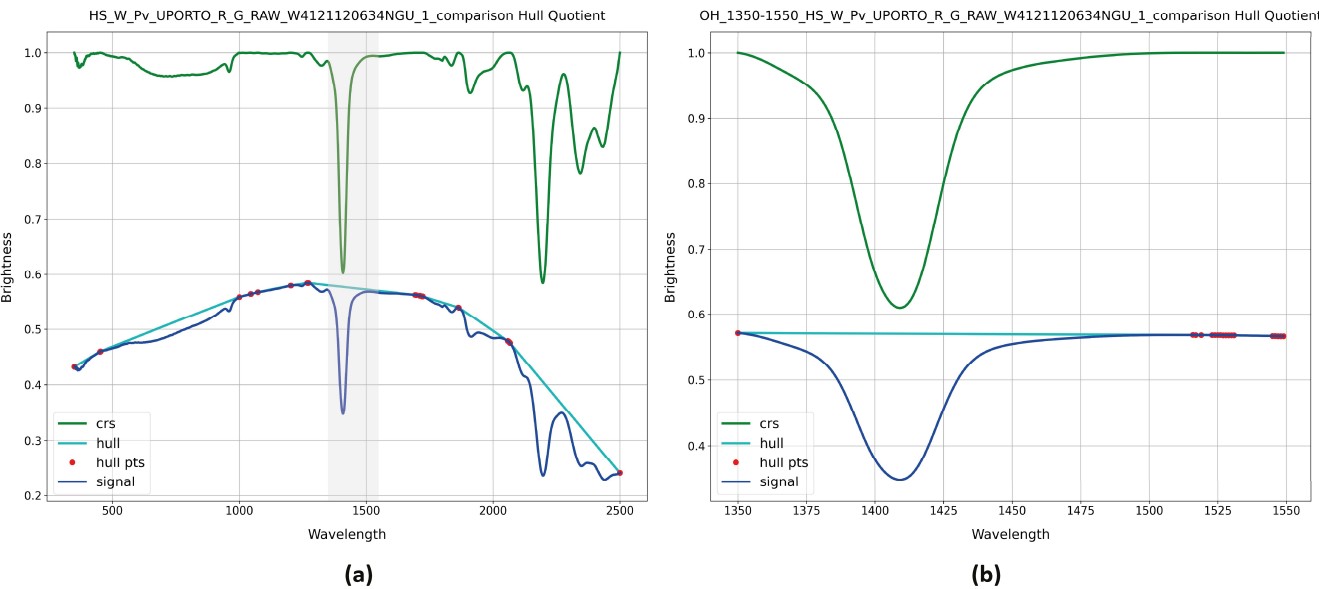

**Figure A2:** Comparison between performing the continuum removal over the entire spectrum (a) and over specific parts of the spectrum (b). In this case, to extract the OH absorption (b), the convex hull was fitted in the 1350-1550 nm range, corresponding to the grey rectangle in Figure (a). crs–continuum removed spectra; pts–points.

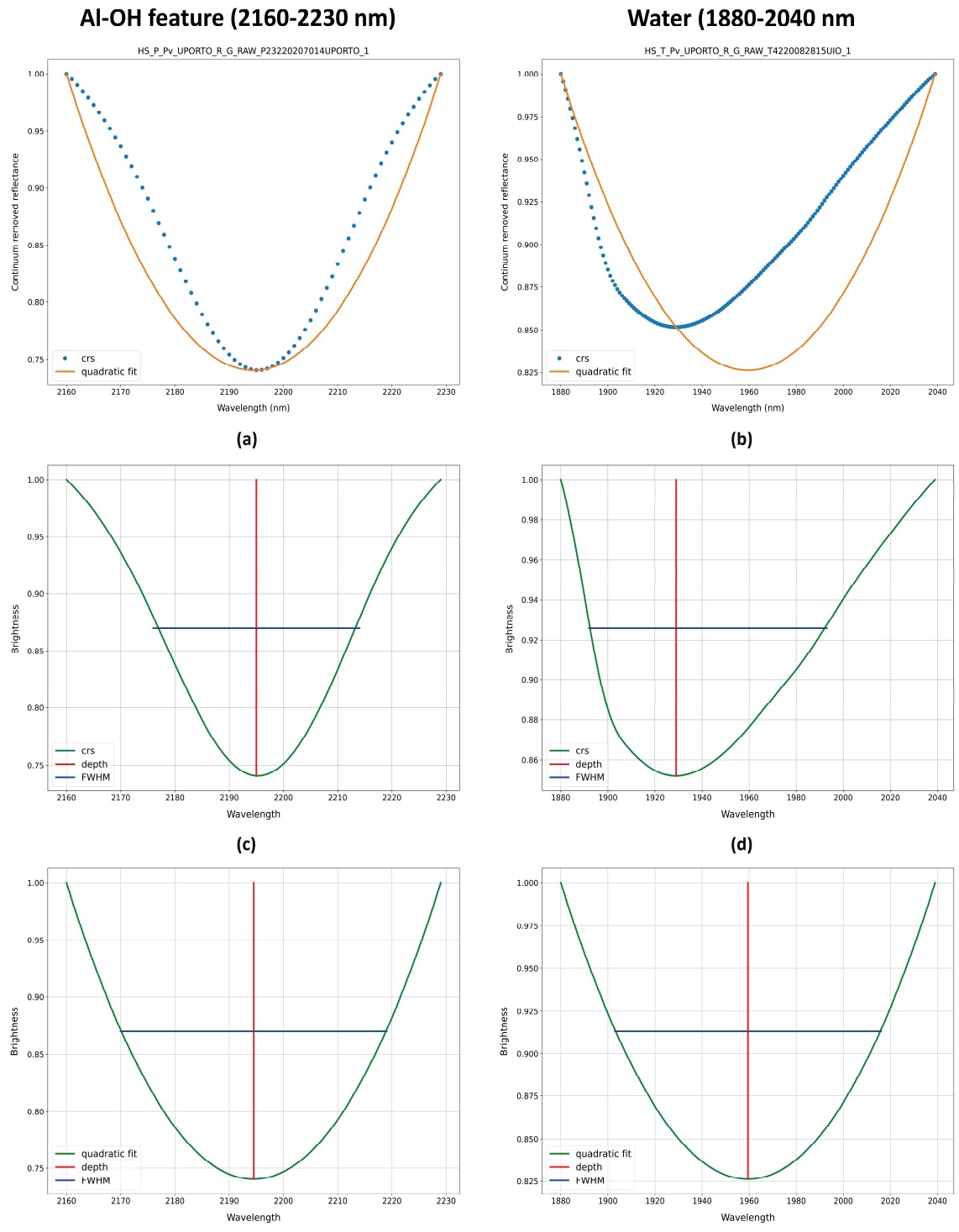

**Figure A3:** Comparison of feature extraction (a-b) using the (c-d) continuum removed spectra (crs) or the (e-f) quadratic function fit; where the two methods show minimal (AlOH feature) or accentuated (water feature) differences. FWHM–full width at half maximum.

**Table A1:** Comparison of feature statistics when employing the continuum-removal or the quadratic function fit methods for feature extraction (Fig. A3). FWHM–full width at half maximum.

| | Al-OH feature (2160-2230 nm) | | Water (1880-2040 nm | |
| --- | --- | --- | --- | --- |
| | Continuum removed | Quadratic function | Continuum removed | Quadratic function |
| Absorption wavelength (nm) | 2195 | 2194.5 | 1929 | 1959.5 |
| Absorption depth | 0.2597 | 0.2597 | 0.1482 | 0.1737 |
| FWHM (nm) | 38 | 49.0482 | 101 | 113.0241 |


**Table A2:** Detailed content of the spectral library.

| Database | Spectra file name | No. spectra/ files | File type |
| --- | --- | --- | --- |
| Tysfjord (Norway) | HS_T_Pv_UPORTO_R_G_RAW_T4220082815UIO_1 and<br>HS_T_Pv_UPORTO_R_G_CR_T4220082815UIO_1<br><br>…<br><br>HS_T_Pv_UPORTO_R_G_RAW_T4221090611NGU_2 and<br>HS_T_Pv_UPORTO_R_G_CR_T4221090611NGU_2 | 131/1309 | |
| Leister (Ireland) | HS_L_Pv_UPORTO_R_G_RAW_L4422012201UCD_1 and<br>HS_L_Pv_UPORTO_R_G_CR_L4422012201UCD_1<br><br>…<br><br>HS_L_Pv_UPORTO_R_G_RAW_L2321102001UPORTO_2<br>and HS_L_Pv_UPORTO_R_G_CR_L2321102001UPORTO_2 | 69/860 | |
| Wolfsberg (Austria) | HS_W_Pv_UPORTO_R_G_RAW_W4121120601NGU_1 and<br>HS_W_Pv_UPORTO_R_G_CR_W4121120601NGU_1<br><br>…<br><br>HS_W_Pv_UPORTO_R_G_RAW_W4121120636NGU_2 and<br>HS_W_Pv_UPORTO_R_G_CR_W4121120636NGU_2 | 76/877 | .png<br>.jpg<br>.txt<br>.pdf<br>.csv |
| Portugal (Adagói, Alijó, Gonçalo) | HS_B_Pv_UPORTO_R_G_RAW_B4220120902UPV_1 and<br>HS_B_Pv_UPORTO_R_G_CR_B4220120902UPV_1<br><br>…<br><br>HS_P_Pv_UPORTO_R_G_RAW_P23220207008UPORTO_3<br>and HS_P_Pv_UPORTO_R_G_CR_P23220207008UPORTO_3 | 34/424 | |
| Spain (Fregeneda) | HS_F_Pv_UPORTO_R_G_RAW_F4220092536UPV_1 and<br>HS_F_Pv_UPORTO_R_G_CR_F4220092536UPV_1<br><br>…<br><br>HS_F_Pv_UPORTO_R_G_RAW_F4220121104UPV_2 and<br>HS_F_Pv_UPORTO_R_G_CR_F4220121104UPV_2 | 18/212 | |

**Team list**

University of Porto: Joana Cardoso-Fernandes, Ana C. Teodoro, Douglas Santos, Cátia Rodrigues de Almeida, Alexandre Lima. University of Oslo: Axel Müller, William Keyser. Geological Survey of Norway: Kerstin Saalmann, Claudia Haase and
Marco Brönner. University College Dublin: Julian Menuge. Geo Unterweissacher GmbH: Ralf Steiner. Universidad del País Vasco-UPV/EHU: Encarnación Roda Robles, Jon Errandonea-Martin and Idoia Garate-Olave.

**Author contribution**

Conceptualization: JC-F, ACT, AL; Data curation: JC-F, DS, CRdA; Formal analysis: JC-F; Funding acquisition: AL; Investigation: JC-F, DS, CRdA, AL; Methodology: JC-F, ACT; Project administration: AL; Resources: JC-F, ACT, AL;
Software: JC-F; Supervision: JC-F, ACT, AL; Validation: JC-F, ACT, DS; Visualization: JC-F; Writing – original draft preparation: JC-F; Writing – review & editing: ACT, DS, CRdA, AL.

**Competing interests**

The authors declare that they have no conflict of interest.

**Acknowledgments**

This study is funded by European Commission's Horizon 2020 innovation programme under grant agreement no. 869274, project GREENPEG New Exploration Tools for European Pegmatite Green-Tech Resources. The GREENPEG project is coordinated by Axel Müller. The Portuguese partners also acknowledge the support provided by Portuguese National Funds through the FCT – Fundação para a Ciência e a Tecnologia, I.P. (Portugal) projects UIDB/04683/2020 and UIDP/04683/2020 (Institute of Earth Sciences). Cátia Rodrigues de Almeida benefits from a Ph.D. grant provided by FCT with reference
PRT/BD/153518/2021. The authors thank European Lithium ECM in Austria and Blackstairs Lithium Limited in Ireland for providing us with samples from their drill cores. Thanks to Axel Müller, Julian Menuge, Kerstin Saalmann, Ralf Steiner, William Keyser, Encarnación Roda Robles, Claudia Haase, and Marco Brönner for collecting and/or sharing the samples.

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
