# Peer review of "Spectral Library of European Pegmatites, Pegmatite Minerals and Pegmatite Host-Rocks – The GREENPEG Project Database"

_Earth System Science Data, 2022_

## Referee Comment (RC2)

Spectroscopy research has been widely applied to remote-sensing exploration and prospecting in recent years and their advantages are highlighted in places where field surveys are difficult to carry out.

The authors shared a spectral database based on the reflectance spectroscopy studies of LCT-, NYF-type pegmatites and host rocks from Austria, Ireland, Norway, Portugal, and Spain. Overall, the dataset is new and can be used to improve the quality of existing representative reflectance spectra. The structure of the data set is well-designed, and the interoperability of the spectral database with a GIS environment is friendly to the users.

However, some descriptions were not clear, and the statements of some important points were inadequate. A major revision is needed before publication.

1. The Python routine proposed in this paper extracts the central wavelength position and the depth of one main absorption feature based on the minimum channel of the observed feature, which may be coarse, since the bandpass of the FieldSpec 4 spectroradiometer is 10nm in SWIR range. A quadratic function fit method was proposed by Raymond Kokaly.

2. As mentioned in the paper, the spectral library contains the spectral mineralogy interpretation as possible, but the details of the technique used to do the spectral mineralogy interpretation are absent.

3. The authors wrote that "Our results show that the spectral mineralogy identified does not necessarily match the minerals identified by observation of hand specimens and optical microscopy. This is because some silicates do not present necessarily diagnostic absorption features (Spatz, 1997) or because the spectra are dominated by alteration minerals that are spectrally very active due to the presence of water/hydroxyl group and superimpose unaltered mineral domains (Line168)". But the pegmatite is highly heterogeneous, the alteration degree of hand specimens in the same pegmatite outcrop probably varies greatly depending on the sampling locations. If these data are used to train the algorithm model, it probably led to incorrect judgment. So, what measures have been taken to ensure the representativeness of the samples?

4. The aim and significance of this spectral database should be more rigorous in the part of introduction and conclusion. The paper mentioned that the database aims to develop tools for the identification of two chemical types of pegmatite (page1, line 25). However, the descriptions in the difference of mineral composition or spectral features in the two types of pegmatite cannot be found. Furthermore, in the conclusion section, it claimed that the spectra data allowed the evaluation of the potential for discriminating both NYF and LCT types with distinct genesis, mineralogy, structure, and host rocks (line 246). On the one hand, because of the signature of $Fe^{2+}$ and $OH^-$, many kinds of minerals such as muscovite, chlorite, illite and montmorillonite could be identified by diagnostic features of spectral reflectance data. On the other hand, it's almost impossible to identify the different types of pegmatite through mineral identification. The database was valuable for industry users, but it is doubted that the aim to identify pegmatite could be achieved by analyzing the spectral features of minerals.

---

## Author Response (AR1)

**Response to Reviewer 1 Comments**

**Thank you for taking the time to review our manuscript and for providing valuable feedback. We appreciate your insightful comments and suggestions, which have helped us improve the quality of our work.**

I'm a little confused as to how the final spectrum of each rock was averaged. The authors say that "each measurement comprises an average of 40 scans with four additional measurements acquired in each analysed spot that were later averaged into a final spectrum." ? - line 138

Did the authors take 40 point measurements on a small area on the surface of each sample (I would assume so)?

That being said, taking averaged spectra from a spot on a pegmatite rock might not be the most scientifically accurate approach. The average specra from that specific area will indicate some mica/clay/ateration mineral at that specific point, this will not be representative of the specific type of pegmatites. How do the authors ensure representativity and completeness?

**Response 1: Thanks for the pertinent question. In fact, several spots are analysed within one sample; for each spot, five measurements are done in a row; each measurement is the automatic average of 40 scans; a final spectrum is created by averaging the five measurements, and the spectra name contains the sample name and the corresponding stop number to discriminate from other spectra acquired in other spots within the same sample. A schematic representation of the spectra acquisition process was added (Figure A1).**

**We agree that taking averaged spectra from a spot on a pegmatite rock is not the most accurate approach. One could argue that a possible approach would be to collect spectra from different areas of the pegmatite rock and then average them to get a more representative spectrum. However, we have tried this approach before in other studies (Cardoso-Fernandes et al., 2020; Cardoso-Fernandes et al., 2021), and noted that some features end up masked by others and some artifacts can be introduced by averaging the spectra. Thus, in this study, we did not judge this as an effective approach since: (i) the actual pegmatite spectrum will not**

match the averaged one; (ii) interpreting the spectral mineralogy using the averaged spectra can lead to missing important spectrally active minerals or to the identification of minerals that are not actually present in the rocks.

We have employed several strategies to ensure the representativity and completeness of the database in this study. First, multiple samples from different parts of each pegmatite (including fresh and weathered regions) and samples from different pegmatites were considered. Drill core samples provided continuous exposure of pegmatite dykes allowing to assess the spatial distribution of mineral assemblages. Taking into account the spatial variability of mineral assemblages within the pegmatite samples, several spots within the samples were measured to obtain representative spectra. Considering the spot size of 10 mm and the variable grain size within pegmatites, it is expected that in coarser-grained areas (pegmatitic texture) individual mineral spectra are obtained, while in fine-grained regions (aplitic texture) the spectra of each spot will represent a rock spectrum of the mineral assemblage within that spot. All results were carefully interpreted in the context of the known mineralogy and geological setting of the area.

We have updated the manuscript to include the aforementioned information (lines 150-154, 173-200, 209-222).

In line 168 the authors say: "Our results show that the spectral mineralogy identified does not necessarily match the minerals identified by observation of hand specimens and optical microscopy. This is because some silicates do not present necessarily diagnostic absorption features (Spatz, 1997) or because the spectra are dominated by alteration minerals that are spectrally very active due to the presence of water/hydroxyl group and superimpose unaltered mineral domains (Hunt and Ashley, 1979)." This statemetn is very true, adding to the issue of representativity.

Response 2: While we acknowledge that the spectral mineralogy identified does not necessarily match the minerals identified by hand specimens, we believe that the use of a spectral library containing a wide variety of mineral types and the collection of spectra from multiple locations within each pegmatite outcrop has allowed us to accurately identify the mineralogy present in our samples. It is noteworthy that some of these clay minerals can dominate the spectra even in the most preserved, fresh samples as demonstrated by Cardoso-Fernandes et al.

**(2021). However, according to previous results, the absorption depth of these alteration minerals appears to correlate with the degree of alteration of the analysed mineral. The manuscript was modified accordingly (lines 66-73, 149-154, 175-180, 209-222, 230-244).**

I disagree with the following statement: Line 172: The representative reflectance spectra stored in the libraries can be utilised for satellite image processing, namely in the image classification tasks. To do so, the acquired spectra can be resampled to match the satellite sensors' spectral resolution and used as a target for algorithm training instead of the image pixels.

Resampling the library spectra (with ~2000 channels) to the spectral resolution of multispectral satellite data (even WV-3; which is 10s of channels) would diminish the characteristic spectral absorption features to such an extent as to make the reference spectrum almost useless for satellite image classification. Large spectral features may be preserved when resampling, however, narrow and sharp diagnostic absorption features can be lost.

**Response 3: We understand the reviewer's concern regarding the potential loss of characteristic spectral absorption features when resampling the library spectra to match the spectral resolution of multispectral satellite data. While we acknowledge the potential limitations of resampling the spectral library to match the spectral resolution of multispectral satellite data, we believe that the benefits of utilizing representative reflectance spectra in image classification tasks outweigh the drawbacks, since resampled spectra can still provide valuable information on important large-scale spectral features relevant for satellite image classification and distinguishing different classes. By training supervised algorithms with resampled library spectra as targets, we can effectively guide the classification process of different outcropping lithologies. We are aware that for mineral identification hyperspectral data is needed. But for lithological mapping, multispectral products can be easily applied as well as demonstrated by several previous works (Asadzadeh and Souza Filho, 2016; Grebby et al., 2011; Rajan Girija and Mayappan, 2019; Rowan et al., 2005).**

**To demonstrate the value of the spectra for satellite image processing, all raw spectra were resampled to correspond to the spectral resolutions of multispectral and hyperspectral sensors, namely Landsat-8 Operational Land Imager (OLI) sensor, Sentinel-2 Multispectral Instrument (MSI) sensor, Worldview-3; and**

**PRISMA. Thus, there is another database level of resampled spectra that the users can download. A new figure (Figure 7) was added to compare the potential discrimination of the pegmatite and host-rock resampled spectra from Wolfsberg. In the case of Landsat-8, Band 6 (1.57-1.65 µm) is crucial for lithological discrimination, with pegmatite showing a reflectance peak in that range that is not observed in the host rocks. The higher spectral resolution of Wordview-3 allows greater discrimination with the AlOH absorption feature well marked for the pegmatite within the SWIR-6 band (2185-2225 nm), while the host rocks only show the carbonate absorption within the SWIR-8 band (2295-2365 nm). The manuscript was modified accordingly (lines 157, 199-203, 292-311).**

**We have also made some changes to the Introduction section to clarify that the reference reflectance spectra can also be very helpful for hyperspectral satellite image processing, especially with the recently launched PRISMA and EnMAP satellites (lines 24-35). Please let us know if you have any further questions or comments regarding this issue.**

Additionally, doing a continuum removal over the entire spectrum can (and in most cases does) create artifacts and distortions. It is recommended to do continuum removal over specific parts of the spectrum the observer is interested in.

**Response 4: We thank the reviewers' input. Yes, it is true that performing the continuum removal on the entire spectrum or just in specific parts of the spectrum can produce different results. At first, for a matter of simplicity, we just extracted multiple features based on the entire spectrum. We have followed the reviewer's recommendation, and performed the continuum removal and extraction of absorption features' statistics based on specific electromagnetic regions where the main absorption features are expected to occur, namely: OH 1350-1550 nm, water 1880-2040 nm, Al-OH 2160-2230 nm, Fe-OH 2230-2296 nm and Mg-OH/CO3 2300-2370 nm. Based on another reviewer's comments, we have also provided the feature statistics based on the quadratic function fit method for all the specific parts of the spectrum. Both, the manuscript (lines 183-200; figures A2-A3) and the database were modified accordingly.**

Furthermore, the authors only present spectra in the VNIR-SWIR range, however, the major rock forming minerals of pegmatites are active in the LWIR range (i.e., quartz and feldspar). This is a major issue, as the authors already showed that the spectra represents clays/micas/alteration minerals, which are not representative of only pegmatites.

**Response 5: Thank you for your comment. We agree that the LWIR range is important for identifying major rock-forming minerals in pegmatites such as quartz and feldspar. However, due to the high cost of LWIR equipment, we were unable to acquire data in this range for our study. We focused on the VNIR-SWIR range, which is more accessible and provides useful information for the identification of clays, micas, and alteration minerals in pegmatites. However, a sensor capable of capturing thermal data would be necessary for a comprehensive characterization of pegmatites and their major rock-forming minerals like quartz and feldspar. Unfortunately, most satellite data products do not include coverage of the thermal region, and if they do, the spatial resolution may be too coarse to be useful for pegmatite identification. Therefore, the spectra contained in the GREENPEG database represent the actual signature that can be captured by the majority of satellite sensors currently available. Users should exercise caution and proper interpretation of these spectra, as it has been established that the spectral mineralogy predominantly associated with pegmatites may not be exclusive to this rock type. While we acknowledge that our study may not provide a complete characterization of pegmatites, we believe that our results can still contribute to the broader understanding of these complex geological formations. The manuscript was modified accordingly (lines 303-311).**

Regarding the database, it is very misleading to show spectra in the SWIR where it indicates some mica or clay but naming the sample K-feldspar. Clearly the spectral measurement does not show feldspar, but probably a SWIR active mineral at that specific point on the rock sample.

**Response 6: We appreciate your observation and would like to address your concern about the potential misrepresentation of spectra in the SWIR region. You are correct in noting that in some instances, the spectra may indicate the presence of mica or clay minerals in the SWIR region, yet the sample is labeled as K-feldspar. The labeling of the samples in the database was done merely by visual inspection and lithological or mineralogical identification based on the hand samples/drill core. However, we acknowledge that at specific points on the rock or mineral sample, other SWIR-active minerals may be present and contribute to the spectral signature captured at those locations. Moreover, only in specific cases can K-feldspar be identified through its SWIR spectra, since the main diagnostic features are in LWIR (Clark et al., 2003; Cardoso-Fernandes et al., 2021). It is important to recognize that the spectra provided in the database serve as representative measurements and not as definitive proof of the presence of a specific mineral throughout the entire sample. For that, other techniques are more suitable such as X-ray diffraction. While we acknowledge that the spectral mineralogy identified does not necessarily match the minerals identified by hand specimens, we believe that the use of a spectral library containing a wide variety of mineral types and the collection of spectra from multiple locations within each pegmatite outcrop has allowed us to accurately identify the mineralogy present in our samples.**

**To address the concern you raised, a clear disclaimer was added in the database documentation to ensure that users understand that the spectra represent dominant mineral signatures in the samples, but localized variations may exist due to the presence of other minerals (lines 209-222). We believe this clarification will help users make more informed interpretations of the spectra in their specific applications.**

All in all, the purpose of this library is unclear. As pegmatites are dominantly composed by quartz, felspars and micas in macro-crystals, it is required to determine a clear sampling strategy. Right now this library shows micas/clays/ ateration minerals and the mixure of those (which again is not unique to pegmatites). What is the purpose of this library? Mapping alterations? REE and Li minerals? Rock forming minerals? As long as this is not clarified and a bespoke sampling strategy adopted, I don't see the utility of such a library.

Response 7: We appreciate the reviewer's concerns. Details on the sampling strategy were provided. And it is noteworthy that in many cases we are dealing with bodies showing varying textures, i.e., aplite-pegmatite dykes. Thus, not only macro-crystals can be analysed, but also aplitic regions that due to their lower grain size are more representative of the overall pegmatite mineral assemblage. We also appreciate the opportunity to further discuss the potential uses and highlight the significance of the spectra library, namely:

- **Pegmatite identification and characterization:** The library serves as a valuable resource for accurately identifying and characterizing pegmatites. Users can leverage the spectra to identify and characterize specific spectrally active minerals within pegmatites. This contributes to a better understanding of pegmatite formation and evolution (alteration).

- **Remote sensing applications:** The spectral signatures in the library can be utilized in remote sensing applications, particularly for satellite image processing and image classification tasks. By resampling acquired spectra to match the spectral resolution of satellite sensors, the library can serve as a valuable training target for algorithms, enabling the automated identification and mapping of pegmatites from satellite imagery. Another approach is the selection of the most useful bands to use in other image processing techniques such as band ratio.

- **Comparative analysis:** The library's comprehensive collection of spectra allows for comparative analysis of pegmatites from different geographic locations.

By providing a comprehensive set of spectral signatures, this library empowers researchers and professionals to better understand, characterize, and map pegmatites, contributing to advancements in geology, mineralogy, and resource exploration. These issues were clarified in the Introduction and Conclusions Sections (lines 64-73, 363-369).

Again, we thank you for your thoughtful review, and we hope that the revised manuscript meets your expectations.

Asadzadeh, S. and Souza Filho, C. R.: A review on spectral processing methods for geological remote sensing, International Journal of Applied Earth Observation and Geoinformation, 47, 69-90, https://doi.org/10.1016/j.jag.2015.12.004, 2016.

Cardoso-Fernandes, J., Silva, J., Lima, A., Teodoro, A. C., Perrotta, M., Cauzid, J., Roda-Robles, E., and Ribeiro, M. A.: Reflectance spectroscopy to validate remote sensing data/algorithms for satellite-based lithium (Li) exploration (Central East Portugal), SPIE Remote Sensing, Earth Resources and Environmental Remote Sensing/GIS Applications XI,  https://doi.org/10.1117/12.2573929,

Cardoso-Fernandes, J., Silva, J., Perrotta, M. M., Lima, A., Teodoro, A. C., Ribeiro, M. A., Dias, F., Barrès, O., Cauzid, J., and Roda-Robles, E.: Interpretation of the Reflectance Spectra of Lithium (Li) Minerals and Pegmatites: A Case Study for Mineralogical and Lithological Identification in the Fregeneda–Almendra Area, Remote Sensing, 13, 3688, 10.3390/rs13183688, 2021.

Clark, R. N., Swayze, G. A., Wise, R., Livo, K. E., Hoefen, T. M., Kokaly, R. F., and Sutley, S. J.: USGS digital spectral library splib05a: https://pubs.usgs.gov/of/2003/ofr-03-395/, last access: 8 January 2021.

Grebby, S., Naden, J., Cunningham, D., and Tansey, K.: Integrating airborne multispectral imagery and airborne LiDAR data for enhanced lithological mapping in vegetated terrain, Remote Sensing of Environment, 115, 214-226, https://doi.org/10.1016/j.rse.2010.08.019, 2011.

Rajan Girija, R. and Mayappan, S.: Mapping of mineral resources and lithological units: a review of remote sensing techniques, International Journal of Image and Data Fusion, 10, 79-106, 10.1080/19479832.2019.1589585, 2019.

Rowan, L. C., Mars, J. C., and Simpson, C. J.: Lithologic mapping of the Mordor, NT, Australia ultramafic complex by using the Advanced Spaceborne Thermal Emission and Reflection Radiometer (ASTER), Remote Sensing of Environment, 99, 105-126, https://doi.org/10.1016/j.rse.2004.11.021, 2005.

**Response to Reviewer 2 Comments**

**We deeply thank the corrections and comments made that will certainly help improve the quality of this manuscript. All considerations were addressed in this new manuscript version. Also, a point-by-point response to each comment are given below.**

1. The Python routine proposed in this paper extracts the central wavelength position and the depth of one main absorption feature based on the minimum channel of the observed feature, which may be coarse since the bandpass of the FieldSpec 4 spectroradiometer is 10nm in the SWIR range. A quadratic function fit method was proposed by Raymond Kokaly.

**Response 1: We thank Reviewer 2 for the suggestion. Absorption feature statistics such as the central wavelength position and the depth of one main absorption feature can be extracted in two ways: (i) in the first, the statistics are computed based on the channel with the minimum value in the continuum-removed feature; (ii) the second relies on the central wavelength of a quadratic function fitted to the band center and one channel on each side. The band center from the quadratic function and the wavelength position of the band center channel may be very close in value when the feature contains a large number of channels. However, the quadratic function is less subject to noise in the spectrum since it uses more than one channel (Kokaly, 2011, 2008). Thus, following the reviewer's advice, we have modified the Python routine to include both statistics (one following the approach of continuum removal and the other based on the quadratic function fit method). Additionally, as advised by Reviewer 1, to avoid the creation of artifacts and distortions, we have applied the continuum removal and respective feature statistics calculation over specific parts of the spectrum where the main absorption features are expected to occur. Both the manuscript (lines 183-200; figures A2-A3; table A1) and the database were modified accordingly.**

1. As mentioned in the paper, the spectral library contains the spectral mineralogy interpretation as possible, but the details of the technique used to do the spectral mineralogy interpretation are absent.

**Response 2: Thank you for the feedback. We apologize for not providing enough details on the technique used for the spectral mineralogy interpretation. However, we have followed the journal's recommendations for data descriptor papers to avoid detailed analysis and extensive interpretations of data while focusing on highlighting the quality, usability, and accessibility of the dataset. Nonetheless, we agree with Reviewer 2 in the sense that details on spectral mineralogy interpretation can be useful for the readers. In short, the mineral identification was performed by comparing the acquired spectra to known reference spectra from the USGS spectral library and other published sources. In addition, we also used spectral feature extraction algorithms to identify specific mineral features in the spectra. The interpretation was done in the continuum-removed spectra by looking at the shape, symmetry, depth and wavelength position of the main absorption features, following the steps proposed by Pontual et al. (2008).**

**We have now updated the manuscript to include more details on the methodology used for spectral mineralogy interpretation, as well as references to relevant literature (lines 209-222, 230-244). Please let us know if you have any further questions or comments.**

1. The authors wrote that "Our results show that the spectral mineralogy identified does not necessarily match the minerals identified by observation of hand specimens and optical microscopy. This is because some silicates do not present necessarily diagnostic absorption features (Spatz, 1997) or because the spectra are dominated by alteration minerals that are spectrally very active due to the presence of water/hydroxyl group and superimpose unaltered mineral domains (Line168)". But the pegmatite is highly heterogeneous, the alteration degree of hand specimens in the same pegmatite outcrop probably varies greatly depending on the sampling locations. If these data are used to train the algorithm model, it probably led to incorrect judgment. So, what measures have been taken to ensure the representativeness of the samples?

**Response 3: Thank you for your comment. We agree that pegmatites are highly heterogeneous and the alteration degree of hand specimens can vary greatly depending on the sampling location. To ensure the representativeness of the samples, we have employed several strategies to ensure the representativity and completeness of the database in this study. First, multiple samples from different parts of each pegmatite (including fresh and weathered regions) and samples from different pegmatites were collected. Drill core samples provided continuous exposure of pegmatite dykes allowing to assess the spatial distribution of mineral assemblages. Taking into account the spatial variability of mineral assemblages within the pegmatite samples, several spots within the samples were measured to obtain representative spectra. Considering the spot size of 10 mm and the variable grain size within pegmatites, it is expected that in coarser-grained areas (pegmatitic texture) individual mineral spectra are obtained, while in fine-grained regions (aplitic texture) the spectra of each spot will represent a rock spectrum of the mineral assemblage within that spot. Thus, the spectral library contains spectra from various mineral types, including both unaltered and altered minerals and fresh/weathered pegmatite samples. All results were carefully interpreted in the context of the known mineralogy and geological setting of the area.**

**While we acknowledge that the spectral mineralogy identified does not necessarily match the minerals identified by hand specimens, we believe that the use of a spectral library containing a wide variety of mineral types and the collection of spectra from multiple locations within each pegmatite outcrop has allowed us to accurately identify the mineralogy present in our samples. We have now updated the manuscript to include the aforementioned information (lines 66-73, 149-154, 173-200, 209-222, 230-244).**

1. The aim and significance of this spectral database should be more rigorous in the part of introduction and conclusion. The paper mentioned that the database aims to develop tools for the identification of two chemical types of pegmatite (page1, line 25). However, the descriptions in the difference of mineral composition or spectral features in the two types of pegmatite cannot be found. Furthermore, in the conclusion section, it claimed that the spectra data allowed the evaluation of the potential for discriminating both NYF and LCT types with distinct genesis, mineralogy, structure, and host rocks (line 246). On the one hand, because of the signature of $Fe^{2+}$ and $OH^-$, many kinds of minerals such as muscovite, chlorite, illite and montmorillonite could be identified by diagnostic features of spectral

reflectance data. On the other hand, it's almost impossible to identify the different types of pegmatite through mineral identification. The database was valuable for industry users, but it is doubted that the aim to identify pegmatite could be achieved by analyzing the spectral features of minerals.

**Response 4: The reviewer raised some valid points regarding the clarity of the paper's objectives and conclusions. In the previous manuscript version, we referred to detailed descriptions of the differences in mineral composition and spectral features between the two types of pegmatite in literature works. We agree that adding details on this subject would help to provide more context for the spectral database and its potential applications. Therefore, Section 1.1. was updated to include more detailed descriptions of the differences between NYF and LCT pegmatites (lines 78-82, 91-94, 100-104, 114-116).**

**In addition, the claim that the spectra data allows for the discrimination of both NYF and LCT types was clarified. We agree that the characteristic signatures of Fe2+ and OH- can be due to different minerals such as muscovite, chlorite, illite, and montmorillonite. However, the overall spectral signature of LCT pegmatites is mostly associated with alteration minerals such as clays while the spectral behaviour of NYF pegmatites is more dominated by biotite/chlorite features. Since biotite and chlorite (to the exception of cookeite) are not expected to appear in LCT pegmatites, the characteristic Fe and Mg features allow to spectrally discriminate between NYF and LCT pegmatites. Despite this, we acknowledge that distinguishing different types of pegmatite solely through spectral features and mineral identification can be challenging. The manuscript was modified accordingly (lines 283-311).**

**Finally, we have clarified that we do not aim to identify the different types of pegmatite through mineral identification. The spectra of pegmatite minerals and rocks can however be used in the decision-making process associated with the selection of the most useful bands for pegmatite identification using remote sensing data and different image processing algorithms.**

**We also appreciate the opportunity to further discuss the potential uses and highlight the significance of the spectra library, namely:**

- **Pegmatite identification and characterization: The library serves as a valuable resource for accurately identifying and characterizing pegmatites. Users can leverage the spectra to identify and characterize**

**specific spectrally active minerals within pegmatites. This contributes to a better understanding of pegmatite formation and evolution (alteration).**

- **Remote sensing applications: The spectral signatures in the library can be utilized in remote sensing applications, particularly for satellite image processing and image classification tasks. By resampling acquired spectra to match the spectral resolution of satellite sensors, the library can serve as a valuable training target for algorithms, enabling the automated identification and mapping of pegmatites from satellite imagery. Another approach is the selection of the most useful bands to use in other image processing techniques such as band ratio.**

- **Comparative analysis: The library's comprehensive collection of spectra allows for comparative analysis of pegmatites from different geographic locations.**

**By providing a comprehensive set of spectral signatures, this library empowers researchers and professionals to better understand, characterize, and map pegmatites, contributing to advancements in geology, mineralogy, and resource exploration. These issues were clarified in the Introduction and Conclusions Chapters (lines 64-73, 363-369).**

Kokaly, R. F.: View_SPECPR: Software for Plotting Spectra (Installation Manual and User's Guide, Version 1.2), Report 2008-1183, 10.3133/ofr20081183, 2008.

Kokaly, R. F.: PRISM: Processing routines in IDL for spectroscopic measurements (installation manual and user's guide, version 1.0), Reston, VA, Report 2011-1155, 10.3133/ofr20111155, 2011.

Pontual, S., Merry, N. J., and Gamson, P.: Spectral interpretation field manual. GMEX Spectral analysis guides for mineral exploration, 3rd, AusSpec International Ltd., Victoria2008.